# Combined Proxies for Heart Rate Variability as a Global Tool to Assess and Monitor Autonomic Dysregulation in Fibromyalgia and Disease-Related Impairments

**DOI:** 10.3390/s25082618

**Published:** 2025-04-21

**Authors:** Emanuella Ladisa, Chiara Abbatantuono, Elena Ammendola, Giusy Tancredi, Marianna Delussi, Giulia Paparella, Livio Clemente, Annalisa Di Dio, Antonio Federici, Marina de Tommaso

**Affiliations:** 1Neurophysiopathology Unit, Department of Translational Biomedicine and Neuroscience (DiBraiN), University of Bari Aldo Moro (IT), 70124 Bari, Italy; emanuella.ladisa@gmail.com (E.L.); chiara.abbatantuono@uniba.it (C.A.); elena.ammendola@policlinico.ba.it (E.A.); tancredigiusy6@gmail.com (G.T.); giulia.paparella@uniba.it (G.P.); livio.clemente@uniba.it (L.C.); annalisa.didio@gmail.com (A.D.D.); 2Department of Education, Psychology, Communication (For.Psi.Com.), University of Bari Aldo Moro (IT), 70124 Bari, Italy; m.delussi@gmail.com; 3School of Medicine, Biomedical Sciences and Human Oncology, University of Bari Aldo Moro (IT), 70124 Bari, Italy; a_federici_it@yahoo.it

**Keywords:** heart rate variability (HRV), correlation dimension (D2), fibromyalgia, autonomic impairment

## Abstract

**Highlights:**

**Abstract:**

Background: Heart rate variability (HRV) provides both linear and nonlinear autonomic proxies that can be informative of health status in fibromyalgia (FM), where sympatho-vagal abnormalities are common. This retrospective observational study aims to: 1. detect differences in correlation dimension (D2) between FM patients and healthy controls (HCs); 2. correlate D2 with standard HRV parameters; 3. correlate the degree of HRV changes using a global composite parameter called HRV grade, derived from three linear indices (SDNN = intervals between normal sinus beats; RMSSD = mean square of successive differences; total power), with FM clinical outcomes; 4. correlate all linear and nonlinear HRV parameters with clinical variables in patients. Methods: N = 85 patients were considered for the analysis and compared to 35 healthy subjects. According to standard diagnostic protocol, they underwent a systematic HRV protocol with a 5-min paced breathing task. Disease duration, pain intensity, mood, sleep, fatigue, and quality of life were assessed. Non-parametric tests for independent samples and pairwise correlations were performed using JMP (all *p* < 0.001). Results: Mann-Whitney U found a significant difference in D2 values between FM patients and HCs (*p* < 0.001). In patients, D2 was associated with all HRV standard indices (all *p* < 0.001) and FM impairment (FIQ = −0.4567; *p* < 0.001). HRV grade was also associated with FM impairment (FIQ = 0.5058; *p* < 0.001). Conclusion: Combining different HRV measurements may help understand the correlates of autonomic dysregulation in FM. Specifically, clinical protocols could benefit from the inclusion and validation of D2 and HRV parameters to target FM severity and related dysautonomia.

## 1. Introduction

Heart rate variability (HRV) is a convenient, non-invasive assessment technique that utilizes the electrocardiographic signal to provide multiple indices of autonomic function through periodic RR intervals [1,2]. It is based on the principle that a denervated heart ideally has a constant and unchanging (i.e., intrinsic) heart rate, such as the hands of a Swiss watch. The nervous system (NS) influences the increase or decrease in heart rate. Thus, the length of the RR intervals reflects the activation of a part of the NS that regulates various visceral functions, i.e., the autonomic nervous system (ANS) [3]. The analysis of different HRV indices allows the assessment of ANS activation and the balance between its sympathetic and parasympathetic components, with an increase in sympathetic activity associated with a decrease in vagal response [4,5]. These indices can be used to investigate the neurophysiological mechanisms underlying the flexibility of cardiorespiratory systems involved in pain and stress responses [6], with interference in sleep, mood, and cognitive disorders [7] and a global impact on the health status of patients [8]. There are several standard HRV parameters used for clinical purposes, some of which have been used in the study of complex chronic pain syndromes. In particular, a reduction in the variability of intervals between normal sinus beats (i.e., SDNN) was recorded very briefly (~30 s), briefly (5 min), or over longer periods of time (up to 24 h) to obtain various interbeat indices is considered predictive of cardiovascular risk, autonomic dysregulation and abnormal pain processing [9,10,11,12]. The root-mean-square of successive differences (i.e., RMSSD) is another metric of parasympathetic cardiac function in the time domain that can be used to investigate neurovisceral self-regulation [13] and also to monitor autonomic impairment in various chronic pain conditions [14]. Furthermore, total HRV power accounts for the overall variance in sympathetic and parasympathetic activity and reflects poorer feedback control in pain patients [15,16]. In addition to the standard HRV parameters, there are other nonlinear measures that can provide information on the random variables of cardio-autonomic dynamics. These measures have not yet been adequately studied in the context of chronic pain syndromes, for which HRV time domain parameters are typically used. These include indices such as the correlation dimension (D2), which can contribute significantly to quantifying the complexity of autonomic function in patients [9]. Indeed, unlike RMSSD or SDNN, which indicate parasympathetic and total variability, D2 estimates the minimum number of variables to construct the system, providing information on the overall predictability of ANS regulatory dynamics [9].

Fibromyalgia (FM) is a chronic pain syndrome with an unclear cause. It is characterized by abnormal autonomic modulation, central sensitization, and, in some cases, peripheral small fiber neuropathy, which contribute to the development and maintenance of somatic FM symptoms [17,18]. The widespread pain sensations with autonomic changes could be caused by abnormal pain processing at both peripheral and central nervous system levels [19]. In recent years, however, interest in the regulatory role of the ANS has increased [20]. This interest has focused on investigating the pathophysiological mechanisms of pain by comparing changes in body temperature, skin conductance, heart rate, and blood flow between healthy individuals and FM patients [21].

Several FM symptoms, such as fatigue, sweating, and bowel problems, appear to be associated with lower autonomic control as patients show increased sympathetic and decreased parasympathetic activity compared to control subjects [22]. Further comparative studies show that HRV indices of vagal modulation are lower in moderate to severe FM regardless of physical activity, indicating abnormal parasympathetic activity in patients under different conditions (rest, during, and after exercise) [23]. Although FM dysautonomia may be associated with lower vagal tone and other signs of autonomic imbalance, it is unclear whether abnormal heart-brain interaction dynamics can be interpreted as a prognostic risk factor for disease severity [2,24].

Integrated assessment and intervention procedures that take into account autonomic imbalance in FM patients could be a useful tool for pain relief and well-being in clinical and home settings [24,25,26].

Considering the current possibilities of using standard and non-standard HRV parameters for clinical and research purposes, the present study aims to (1) detect differences in D2 values in FM patients compared to healthy controls to assess whether the complexity of autonomic nonlinear dynamics varies in the clinical population; (2) correlate D2 with HRV standard parameters to understand whether there are correspondences between nonlinear and linear measures of HRV in FM patients; (3) correlate the degree of HRV changes by a global composite HRV parameter consisting of three linear indices (SDNN, RMSSD, total power) with clinical outcomes of FM; (4) correlate all linear and nonlinear HRV indices with clinical variables in patients to gain a better knowledge of the assessment and treatment of FM from a multidimensional perspective.

## 2. Materials and Methods

### 2.1. Subjects

In this retrospective observational study, we analyzed data from 85 consecutive patients who came to our neuropathic pain center between February 2023 and February 2024 and were diagnosed as FM according to the American College of Rheumatology inclusion criteria [18]. Normative data were collected among university and hospital staff (80 cases in total), so 39 healthy age- and sex-matched subjects served as control (HCs; F = 32, M = 7; mean age = 47.7 ± 12.3 years; range = 25–68 years). Exclusion criteria for all participants were cardiac arrhythmias, including benign arrhythmias (e.g., extrasystoles) interfering with the overall interpretation of the data, general medical and metabolic disease, a diagnosis of neurologic disease, with the exclusion of primary headache, use of pain medication in the 24 h prior to the neurophysiological examination, current use of CNS-active medications. All patients are advised to start pharmacological treatment after the neurophysiological examination. Before participating in the study, subjects were instructed to abstain from coffee, smoking, and other stimulant substances for at least 3 h before the recording and were given instructions on how to perform the test. The clinical and neurophysiological examination was conducted following a routine clinical practice for patients with FM [17].

Clinical data were recorded through an electronic database https://neuroclinic.thcs.it/login/login.html, accessed on 31 January 2016) authorized by our Ethical Committee. Prior to the examination, patients signed an informed consent for their clinical data insertion into the electronic database and the use of clinical and neurophysiological data for research purposes.

### 2.2. Neurophysiological Assessment

Polygraphic data were acquired, visually inspected, and selected by Biopac MP100 apparatus and Acqknowledge 4.1 software (Biopac Systems Inc., CA, USA). HRV assessment in the selected polygraphic segments was made by Kubios 2.0 software (Kubios Oy, Finland).

In a quiet room, the participants were comfortably laying on an armchair in a semi-reclining position while their electrocardiographic (ECG) waveform was recorded by three electrodes attached to the left wrist, right wrist, and right clavicle. Two additional electrodes were placed on the right hand for microconductance recording, and a Doppler probe (Biopac MP100 apparatus) was positioned to monitor peripheral blood flow. The last two recording channels were not used for direct analysis but were used to provide feedback to the technical staff on the patient’s state of relaxation and to indicate any changes in the patient’s state of activation due to external disturbances. After preliminary adaptation to the environment, the recording sessions lasted 10 min. Artifact-free data were subsequently analyzed.

To assess HRV, a polygraphic segment lasting 5 min was successively selected in each recording session (i.e., brief HRV monitoring) on the basis of stability of recorded parameters. The sequence of numerical values of the R-R intervals duration was extracted from each 5-min segment, and each R-R sequence was processed by Kubios. In particular, the values of SDNN, RMSSD, and D2 were calculated using appropriate formulas to form vectors from the time series RR, compute the number of vectors, and then determine the correlation dimension (https://www.kubios.com/blog/hrv-analysis-methods/, accessed on 20 April 2025). In the software, an embedding default value of m = 10 was selected.

In addition to the separate HRV indices that were collected, a summary grade was derived from SDNN, RMSSD, and total power to obtain a score of standard HRV parameters. This tool, set as a graded scale, was developed to complement standard assessment, yielding a summary to screen for FM patients based on their overall ANS functioning. HRV grade was measured on a 4-point scale ranging from 0 = no abnormalities to 3 = abnormalities on all three HRV indices. The presence or absence of abnormalities was based on short-term ECG norms reported in the literature. In particular, abnormalities were assessed by comparison with the extensive review of HRV normal values by Nunan et al. [27], summarized by Shaffer and Ginsberg [9]. This further measurement was useful to investigate whether D2 could be consistent with multiple linear indices of HRV and whether clinical data could be related to this composite score.

### 2.3. Clinical Assessment

The Neuroclinic electronic database (https://neuroclinic.thcs.it/home.php, accessed on 20 April 2025) includes the Italian version of the Fibromyalgia Impact Questionnaire (FIQ) [28] to assess the impact of FM symptoms on participants assigned to the FM clinical group. All FIQ items refer to the past week and can be scored at different levels of impairment, with a maximum raw score of 100. Anxiety and depression symptoms are measured using the Self-rating Anxiety Scale—SAS [29] and the Self-rating Depression Scale—SDS [30]. The Short-form Health Survey (SF-36), a questionnaire with 36 items, is also included in the Neuroclinic database to assess the perception of quality of life [31]. The Multidimensional Assessment of Fatigue (MAF) is used to assess different dimensions of fatigue experienced in the last 7 days, with a maximum score of 50 [32]. The assessment of sleep quality from the Medical Outcomes Study (MOS) is also included in the protocol [33]. In assessing pain, the Numerical Rating Scale (NRS) is applied to rate self-reported pain on an intensity scale from 0 to 10. Clinical measures of the Widespread Pain Index (WPI) and the Symptom Severity (SS) Scale serve to confirm FM diagnosis [18].

### 2.4. Statistical Analysis

To compare D2 between the FM patients and the HCs, we previously tested the normality hypothesis with Shapiro-Wilk (W = 0.899; *p* < 0.001). As the normality hypothesis was rejected, the non-parametric Mann-Whitney U was applied. The achieved power was 91.37% (α = 0.05, two-tailed; n1 = n2; independent samples on which mean and SD of D2 were measured; further details are reported in Table 1) according to post-hoc analysis. Multivariate analysis between clinical and neurophysiological variables was performed in the FM group by calculating Pearson’s and Spearman’s correlation coefficients. Age was included as a covariate as it may influence cardiac variability. The JMP 17.2.0 software was used. We considered correlations with a significance < 0.001.

## 3. Results

Of the eligible subjects, 4 patients were excluded for cardiac rhythm abnormalities, 5 patients were under CNS-acting drugs at their first visit, 1 FM patient was also not considered for HRV analysis due to technical problems during recording, so 85 FM patients (mean age = 50.8 ± 11.8 years; range = 26–72 years; 68 F, 17 M) were included for the analysis and compared with the age- and sex-matched control subjects (n = 35).

For each group, i.e., HC and FM, the mean, median, standard deviation, variance, and asymmetry with respect to D2 were calculated (Table 1).

The non-parametric Mann-Whitney U showed a significant D2 difference between the FM patients and the controls (Mann-Whitney U 1013 *p* < 0.001) (Figure 1).

With regard to FM patients, the multivariate analysis showed significant correlations between standard and non-standard HRV parameters, and the clinical variables collected. Details about multivariate analysis are reported in Appendix A (Appendix A). D2 was found to be positively associated with all HRV indices except the LF/HF ratio. In particular, D2 was significantly associated with the RR-Tri index, as shown by the Spearman correlation coefficients (0.8420), NN50 (0.7619), pNN50 (0.7363), SD2 (0.7014), SDNN (0.6213), LF (0.5991), HF (0.5705), total power (0.5702), VLF (0.4903), and RMSSD (0.4517) (all *p* < 0.001).

The overall level of HRV was associated with the FIQ (0.5058; *p* < 0.001).

FM impairment was also negatively associated with D2 (D2 = −0.4567; *p* < 0.001). In addition, FM impairment, as assessed by the FIQ, was significantly associated with perceived pain (NRS = 0.6976; *p* < 0.001), physical health (SF-36-PHI = −0.5720; *p* < 0.001) (Figure 2).

## 4. Discussion

The results of our study confirmed a reduction in cardiac variability in patients with fibromyalgia with significant associations between heart rate fluctuations by D2 and standard HRV indices in the time domain. Overall, this pilot study also demonstrated the correlation between D2, a novel composite index (i.e., HRV grade) derived from three linear HRV parameters (i.e., SDNN, RMSSD, total power), FM impairment, and perceived pain.

A first innovative aspect of this work lies in the anomalies of D2 values in patients compared to control subjects, as an additional parameter in the time domain. D2 indicates the complexity of heart rate patterns, which in turn may reflect neuronal oscillations and functional network properties of the brain [34]. The limited use of D2 and other nonlinear HRV parameters in clinical settings may primarily depend on the lack of standardization. In addition, the difficulty of translating mathematical measures of HRV complexity into a more intuitive neurophysiological and clinical language may represent a further challenge to D2 adoption in clinical settings. This points to the number of dynamic variables needed to define the broader framework of HRV time series and requires further research on their standardization in the context of specific clinical conditions and associated impairments [6]. In the last decade, evidence has been found in the literature for the versatile applications of D2 in different conditions [35,36,37,38]. In our work, D2 was studied along with other frequency domain measurements to understand the vagal and sympathetic ANS responses in FM. This provided the dual opportunity to determine whether the complexity of heart rate fluctuation differs between FM patients and HCs and to elucidate the relationships between D2, standard HRV indices (including a comprehensive HRV grade), and clinical outcomes in FM. Since D2 differs significantly between FM and HC groups, it shows autonomic dysregulation in FM also in terms of the complexity of heart rate patterns. This result could contribute to the understanding of the basal neurogenic mechanisms of FM and confirm the hypothesis of ANS involvement in this disease, including proxies for its (non-)linearity.

From correlation analysis, we also found a moderate association between D2 and an index of FM-specific impairment (FIQ). Specifically, D2 represents a nonlinear measure of system-wide complexity rather than a marker of sympathetic overactivity, while the FIQ is a self-report measure that globally assesses FM impact on a series of tasks and social activities that are usually performed on a weekly basis. The negative correlation between these two variables may be interpreted in view of patients’ chronic difficulties in modulating physiological responses to pain and coping with environmental challenges [39]. Maladaptive homeostatic equilibrium in chronic pain was discussed in a recent study where an association between disease-related stress and long-term rearrangements in cardiovascular (re)activity emerged in a sample of FM patients [39]. In addition, we found significant associations between D2 and HRV linear indices (i.e., RR-Tri index, NN50, pNN50, SD2, SDNN, LF, HF, total power, VLF, and RMSSD). In particular, D2 seems to be associated with parasympathetic-dominant parameters that may result in altered chronic pain. However, among all HRV indices considered individually, D2 was the most closely linked with FM-specific functional impairment, and this finding may suggest the utility of complementing FM standard assessment using HRV system-level measures that could reflect patients’ impairment in daily life and foster intervention targeting autonomic plasticity and resilience in FM.

This approach could bridge the gap between mathematical constructs and clinically relevant parameters of FM and support the applicability of D2 in the clinical context of chronic pain syndromes. With regard to the neurophysiological and clinical meaning of the D2 value, we can suppose that it could reflect the activity of oscillators generating the rhythms we can find in HRV. The D2 value seems to reflect the number of main spectral components of HRV. In the FM patients, both a decrease in D2 values and a decrease in total and single bands power of Fourier spectra were found. This could suggest that in FM patients, the ability to generate collective synergetic patterns of regulatory rhythmical activity is poorer than in control subjects. Further studies are, however, required for a better understanding of the clinical meaning of these results [40].

With regard to our results, the four-level HRV indices showed a correlation with the main clinical features, including FM disability. Indeed, a comprehensive HRV score, which is non-invasive and painless for the patient, can be used as an indirect biomarker of the impact of FM on daily life as it reflects the extent of ANS involvement. This result appears consistent with the association between FM impairment and D2 as an integrated index for autonomic flexibility and could also suggest that the inclusion of HRV grade as a potential tool for risk assessment, monitoring disease progression, or response to treatment, may be particularly beneficial for the clinical interpretation of HRV scores in pain conditions characterized by multiple autonomic changes. An additional implication when considering the magnitude of HRV change in relation to the degree of impairment caused by the disease is that patients with higher HRV also tend to report better quality of life [41], as HRV is assessed using a method for which there is an extensive literature on its applications. Factors such as sleep and autonomic functions may play a key role in mediating stress exposure and improving quality of life in FM due to their involvement in self-regulation [42,43]. In addition, the complex interplay between central and peripheral changes could lead to the ANS abnormalities detected with the present results [19,44]. The study of both central and peripheral nervous system dysfunction in fibromyalgia may benefit from the evidence of imbalanced sympathetic/parasympathetic responses [45]. This suggests that targeted interventions to improve autonomic balance could have a significant impact on the perceived quality of life in this population. It would be beneficial to determine whether and to what extent improvements in heart rate variability (HRV) could be associated with the perception of the disease or its symptoms.

Despite these promising results, the available data have significant limitations, including the relatively small sample size and methodological variability in the measurement of the various HRV parameters, especially when non-standardized ones are involved. In this regard, system-wise measures such as D2 and global measures such as HRV grade could complement and not replace standardized linear indices that are informative of branch-specific ANS dysfunctions in FM. Another limitation of the study concerns the retrospective nature of the chosen research design, which does not allow us to understand longitudinal changes in the flexibility of the neurovegetative system over time. It is also necessary to complement the data obtained by HRV and FIQ with other clinical and psychological variables (e.g., cognitive and emotional function, medication, etc.) to ensure robust clinical applicability of our results. The causes of ANS imbalance should be clarified in individual patients to determine the role of small fiber involvement by skin biopsy and neurophysiological assessment [17].

## 5. Conclusions

This retrospective observational study sought to (1) detect differences in D2 between FM patients and HCs; (2) correlate D2 with standard HRV parameters; (3) correlate HRV grade with FM clinical outcomes; (4) correlate all linear and nonlinear HRV parameters with clinical variables in patients. D2, which differed significantly between patients and HCs, indicates reduced ANS complexity and flexibility, reflecting lower adaptability in chronic pain conditions such as FM. This parameter also emerged as significantly associated with FM impairment and linear HRV measures. In this framework, reduced D2 and altered HRV linear parameters may indicate not only decreased variability in FM but also less information processing capacity to internal or environmental demands. Even if D2 still requires standardization, its inclusion of D2 in HRV protocols to complement linear indices could be a promising addition to FM clinical assessment to provide an index of autonomic volitional impairment. Likewise, HRV grade was associated with functional impairment in FM. The use of a graded score of linear HRV parameters as an additional screening tool could contribute to a first-level stratification of FM patients based on different neurovegetative features. Being a composite index to be validated, the HRV grade could intuitively and synthetically suggest overall functioning in FM by quantifying the grade of impairment of the vegetative system in clinical practice. Although our results are preliminary, they may support the utility of nonlinear HRV measures and a composite score from linear HRV parameters to detect dysfunction of the cardio-autonomic interplay and related impairment in FM. Further research with a larger and more balanced sample size could focus on non-pharmacological strategies aimed at normalizing HRV parameters and possibly clinical features, such as biofeedback techniques, mindfulness meditation, and physical exercise.

## Figures and Tables

**Figure 1 sensors-25-02618-f001:**
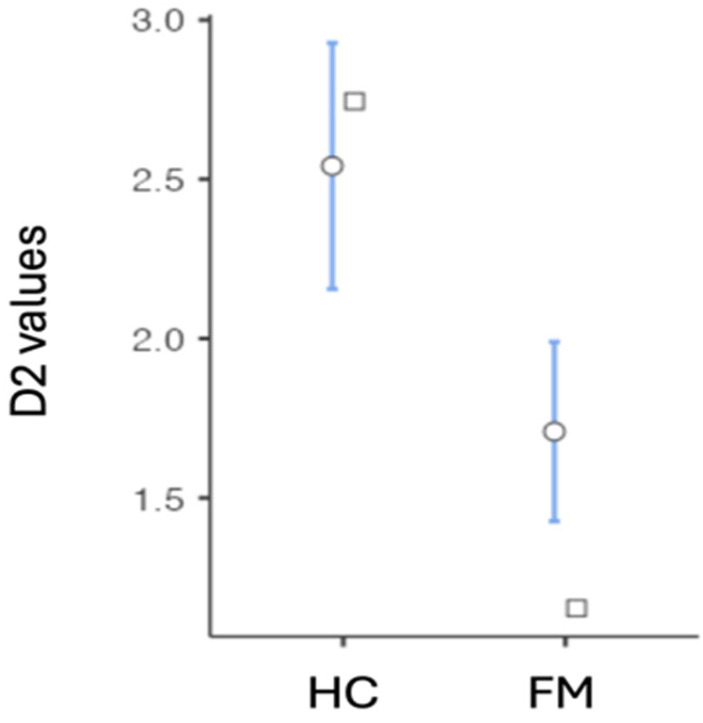
Differences in correlation dimension (D2) between healthy controls and fibromyalgia patients. Acronyms: D2 = correlation dimension of Heart Rate Variability; FM = Fibromyalgia group; HC = Healthy control group. The round-shaped symbol indicates the mean (confidence interval = 95%). The square-shaped symbol indicates the median.

**Figure 2 sensors-25-02618-f002:**
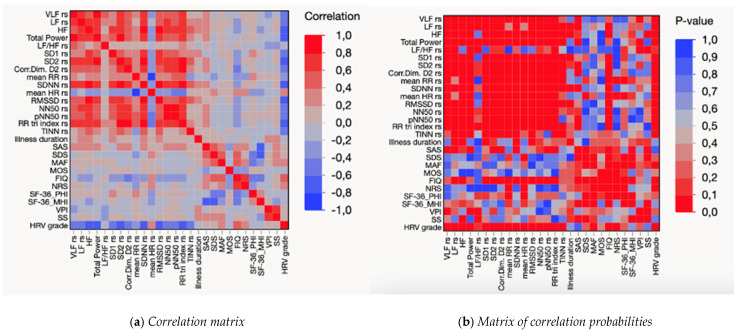
Matrices of correlations (**a**) and correlation probabilities (**b**) between indices of heart rate variability and clinical variables in FM patients. Notes: control for age; main acronyms are listed in the abbreviation section, whereas the full, detailed table (Appendix A is attached as Appendix A. As data did not distribute normally, Spearman’s pairwise correlation was used for the analyses (Appendix A), whereas Pearson was used for further testing in addition to non-parametric statistics (Figure 2).

**Table 1 sensors-25-02618-t001:** Descriptive statistics computed for correlation dimension of Heart Rate Variability in healthy controls and fibromyalgia patients.

Group		N	Mean	Median	SD	SE	Variance	Min.	Max.
D2	HC	39	2.54	2.75	1.23	0.197	1.51	0.642	4.55
	FM	85	1.71	1.15	1.33	1.33	1.76	0.00	3.81

Notes: Acronyms: D2 = correlation dimension of Heart Rate Variability; FM = Fibromyalgia group; HC = Healthy control group; N = Sample numerosity; SD = Standard deviation; SE = Standard error.

## Data Availability

Data used to support the findings of this study are available from the corresponding author upon request.

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
