# Peer review of "Combined Proxies for Heart Rate Variability as a Global Tool to Assess and Monitor Autonomic Dysregulation in Fibromyalgia and Disease-Related Impairments"

_sensors, 2025, doi:10.3390/s25082618_

Round 1

Reviewer 1 Report

Comments and Suggestions for Authors

Thank you for the opportunity to review the manuscript titled, “Combined proxies for heart rate variability as a new tool to assess and monitor autonomic dysregulation in fibromyalgia and disease-related impairments.” The manuscript was well written in general.

Introduction

Line 77: Author(s) propose that D2 can add or contribute to understanding ANS function but fail to further describe this analysis. Since D2 analysis is the first aim (see Line 102) it leaves the reader asking questions on what this analysis is and how its consideration adds to the understanding of ANS function. The discussion (starting on Line 245 does provide some information on D2 analysis. I suggest 1-2 sentences around Line 77 that adds a bit more on D2 to set up the first aim and leads into your discussion section.

Materials & Methods

Line 113: Reads, “ In this observational study, we analyzed retrospectively data from …” Simply state this is a retrospective observational study design – or something like that. The way it reads now is not as clear and the transition is not optimal for the general reader.

In the Abstract (Line 28) also make this change for research design clarity (i.e., “This retrospective observational study …”).

Line 113-114: Remove one of the “consecutive” words.

Line 138: You had the participants sitting. Why not lying supine? Is there a difference between positions for HRV recordings? Provide your rational for why you chose to collect the data sitting vs. supine? I believe sitting is fine but would like to know your rational either here or in the Limitations Section.

Line 146: This statement is not clear, “… the recording sessions lasted until 10 minutes at least of artifact-free data…” Reword this for clarity.

Line 151: Was the D2 integer or fractal values? Explain which one and briefly explain the difference between them and what that means for the reader.

Lines 173-181: Please provide the reader with a better understanding of the “summary grade” and the “composite” score. Add to what is already here to provide clarity and a better understanding of what this is and how it adds to your assessment.

Line 198: What was your prospective power analysis? This is where it is confusing in regards to observational vs. retrospective study design.

Line 199: “39 HC” – Why not the same number (e.g., 85) of matched controls?

Line 208: Table 1 provides D2 values. Are there a set of standard expected D2 values?

Discussion

Line 256: Reads, “…used in clinical practice may be due difficulty of…”. Please edit this for clarity [e.g., “may be due difficulty”].

**In the Abstract (Lines 28-34) the authors provide 4 aims to their study. In the “Results” section, I would like to see the authors clearly link the analysis for each aim to improve readability. Then, in the “Conclusions” the authors can provide their interpretation of their aims based on their analysis and interpretation. That would make this clearer.

**I did not see the “Supplementary materials” file in the submitted information. I would like to review this material.

Author Response

Reviewer1

R1: Thank you for the opportunity to review the manuscript titled, “Combined proxies for heart rate variability as a new tool to assess and monitor autonomic dysregulation in fibromyalgia and disease-related impairments.” The manuscript was well written in general.

Line 77: Author(s) propose that D2 can add or contribute to understanding ANS function but fail to further describe this analysis. Since D2 analysis is the first aim (see Line 102) it leaves the reader asking questions on what this analysis is and how its consideration adds to the understanding of ANS function. The discussion (starting on Line 245 does provide some information on D2 analysis. I suggest 1-2 sentences around Line 77 that adds a bit more on D2 to set up the first aim and leads into your discussion section.

We thank Reviewer1 for this feedback. We added one sentences (lines 80-82) to better explain D2 meaning: “Indeed, unlike RMSSD or SDNN that indicate parasympathetic and total variability, D2 estimates the minimum number of variables to construct the system, providing information on the overall predictability of ANS regulatory dynamics [9].” Consistent with other Reviewers’ suggestions, we also extended the conclusive remarks in our work to summarize D2 applications and its potential implications in the context of FM neurophysiological assessment (lines 330-336): “This parameter also emerged as significantly associated with FM impairment and linear HRV measures. In this framework, reduced D2 and altered HRV linear parameters may indicate not only decreased variability in FM, but also less information processing capacity to internal or environmental demands. Even if D2 still requires standardization, its inclusion of D2 in HRV protocols to complement linear indices could be a promising addition to FM clinical assessment to provide an index of autonomic volitional impairment.”

R1: Materials & Methods

Line 113: Reads, “In this observational study, we analyzed retrospectively data from …” Simply state this is a retrospective observational study design – or something like that. The way it reads now is not as clear and the transition is not optimal for the general reader.

In the Abstract (Line 28) also make this change for research design clarity (i.e., “This retrospective observational study …”).

We enhanced clarity and readability, and made the suggested changes (lines 28; 116).

R1: Lines113-114: Remove one of the “consecutive” words.

The additional word was a typo and thus was removed.

R1: Line 138: You had the participants sitting. Why not lying supine? Is there a difference between positions for HRV recordings? Provide your rational for why you chose to collect the data sitting vs. supine? I believe sitting is fine but would like to know your rational either here or in the Limitations Section.

We thank Reviewer1 for this important comment and specify that the chair we used was almost fully reclinable so that participants could assume a (supine) resting position, although not fully extended as the part where the head lies was slightly uplifted. Consistently, we specified that the actual position of participants during HRV recording was semi-reclined. We also changed the verb “sitting” using “laying on” as it was more appropriate.

R1: Line 146: This statement is not clear, “… the recording sessions lasted until 10 minutes at least of artifact-free data…” Reword this for clarity.

We edited the sentence as follows (lines 148-149): “After preliminary adaptation to the environment, the recording sessions lasted 10 minutes. Artifact-free data were subsequently analyzed.”

R1: Line 151: Was the D2 integer or fractal values? Explain which one and briefly explain the difference between them and what that means for the reader.

We thank Reviewer1 for this observation. D2 values reported in our study are real numbers with decimal components (e.g., Table 1). D2 computed via Kubios estimates the attractor dimension of the physiological system based on a fractal model:

https://www.kubios.com/blog/hrv-analysis-methods/

As for the comparative analysis between FM patients and HCs, its parameters are expressed in form of noninteger values. Higher D2 indicates greater complexity of the cardiac system (typically associated with better physiological adaptation), whereas lower D2 may suggest reduced variability and more rigid dynamics (e.g., under stress conditions or disease).

  1. Guzzetti, M.G. Signorini, C. Cogliati, S. Mezzetti, A. Porta, S. Cerutti, and A. Malliani. Non-linear dynamics and chaotic indices in heart rate variability of normal subjects and heart-transplanted patients. Cardiovascular Research, 31:441–446, 1996.
  2. Henry, N. Lovell, and F. Camacho. Nonlinear dynamics time series analysis. In M. Akay, editor, Nonlinear Biomedical Signal Processing: Dynamic Analysis and Modeling, volume II, chapter 1, pages 1–39. IEEE Press, New York, 2001.

R1: Lines 173-181: Please provide the reader with a better understanding of the “summary grade” and the “composite” score. Add to what is already here to provide clarity and a better understanding of what this is and how it adds to your assessment.

We thank Reviewer1 for this key point. We added the following explanatory sentence (lines 160-161) to better explain our choice to include HRV grade as a potential screening tool in FM: “This tool, set as a graded scale, was developed to complement standard assessment, yielding a summary to screen for FM patients based on their overall ANS functioning.”

R1: Line 198: What was your prospective power analysis? This is where it is confusing in regards to observational vs. retrospective study design.

We appreciate Reviewer1’s insightful comment regarding power analysis. We confirm our study is retrospective in nature. Accordingly, we edited section 2.4 Statistical Analysis specifying that we used a post-hoc approach (lines 187-189). As per the attached figure, we used GPower* to preliminary calculate Cohen’s d from the observed group differences, and to calculate the achieved power. Analysis confirmed that our study achieved adequate power (>.91) to detect the significant differences between groups.

R1: Line 199: “39 HC” – Why not the same number (e.g., 85) of matched controls?

Given the retrospective nature of the study, our controls were a set of healthy subjects to obtain laboratory normative data for clinical purposes. Output from post-hoc analysis was satisfying, as the study achieved adequate power (>.91).

R1: Line 208: Table 1 provides D2 values. Are there a set of standard expected D2 values?

We mentioned in the introduction and discussions that D2 is not a value commonly used in clinical practice and currently lacks clinical normative data. For this reason, we considered it important to preliminarily investigate whether this value might differ from matched HCs.

R1: Discussion

Line 256: Reads, “…used in clinical practice may be due difficulty of…”. Please edit this for clarity [e.g., “may be due difficulty”].

We thank Reviewer1 for this suggestion. During this first round of revisions, following the suggestion of another Reviewer, we changed the full sentence to give a broader perspective on this topic (lines 243-247).

R1: **In the Abstract (Lines 28-34) the authors provide 4 aims to their study. In the “Results” section, I would like to see the authors clearly link the analysis for each aim to improve readability. Then, in the “Conclusions” the authors can provide their interpretation of their aims based on their analysis and interpretation. That would make this clearer.

We acknowledge the results were reported synthetically in the manuscript, however, we ensured they could follow the same structure of the study aims, starting from D2 comparison between FM patients and controls (1st result – 1st study aim; a significant difference was found between samples), followed by correlations between D2 and HRV linear parameters (2nd result – 2nd study aim; significant associations emerged between D2 and multiple HRV linear parameters), correlations between HRV grade and FM outcomes (3rd result – 3rd study aim; significant associations emerged between HRV grade and FIQ) and (4th result – 4th study aim; a significant association emerged between FIQ and D2). The change we made in the order was marked in yellow (line 221). We also made changes in the Conclusions (lines 324-347) to ensure better clarity and understanding.

R1: **I did not see the “Supplementary materials” file in the submitted information. I would like to review this material.

We noticed Table 3 was missing from the version of the manuscript under revision. In case supplementary materials cannot be downloaded from the link provided in the proper section of the manuscript, it should also be visible now as we embedded it within the body of text, following the References (Table 3).

Reviewer 2 Report

Comments and Suggestions for Authors

The aim of the study was to prove the significance of the D2 assessment for characterizing autonomic dysfunction in patients with fibromyalgia. The present study aims to: (1) detect differences in D2 values in FM patients compared to healthy controls to assess whether the complexity of autonomic nonlinear dynamics varies in the clinical population; (2) correlate D2 with HRV standard parameters to understand whether there are correspondences between nonlinear and linear measures of HRV in FM patients; (3) to correlate the degree of HRV changes by a new composite HRV parameter consisting of three linear indices (SDNN, RMSSD, total power) with clinical outcomes of FM; (4) to correlate all linear and nonlinear HRV indices with clinical variables in patients to gain a better knowledge of the assessment and treatment of FM from a multidimensional perspective.

The search for HRV indices that are reliable for characterizing autonomic disorders is important. However, it should be taken into account that all HRV parameters are based on a common primary parameter, i.e., the RR interval, the inverse of the HR. Thus, a high cross-correlation of HRV indicators was revealed, which is not of high value and may produce a cyclic error.

It is advisable to select a limited number of HRV parameters obtained by different methods of analysis and reflecting the contribution of different regulatory mechanisms. It is more valuable to identify how a comprehensive HRV assessment reflects the degree of autonomic disorders and correlates with the severity of the disease.

The authors used the nonlinear index D2 of the HRV, which differed in patients with fibromyalgia and healthy individuals. At the same time, the studied parameter does not show the structure of autonomic disorders in patients with fibromyalgia. The authors should make a more in-depth analysis showing the structure of autonomic disorders in fibromyalgia.

Conclusion: the article needs revision, a more in-depth analysis of the HRV data for accurate clinical interpretation.

Author Response

Reviewer2

R2: The aim of the study was to prove the significance of the D2 assessment for characterizing autonomic dysfunction in patients with fibromyalgia. The present study aims to: (1) detect differences in D2 values in FM patients compared to healthy controls to assess whether the complexity of autonomic nonlinear dynamics varies in the clinical population; (2) correlate D2 with HRV standard parameters to understand whether there are correspondences between nonlinear and linear measures of HRV in FM patients; (3) to correlate the degree of HRV changes by a new composite HRV parameter consisting of three linear indices (SDNN, RMSSD, total power) with clinical outcomes of FM; (4) to correlate all linear and nonlinear HRV indices with clinical variables in patients to gain a better knowledge of the assessment and treatment of FM from a multidimensional perspective.

The search for HRV indices that are reliable for characterizing autonomic disorders is important. However, it should be taken into account that all HRV parameters are based on a common primary parameter, i.e., the RR interval, the inverse of the HR. Thus, a high cross-correlation of HRV indicators was revealed, which is not of high value and may produce a cyclic error.

It is advisable to select a limited number of HRV parameters obtained by different methods of analysis and reflecting the contribution of different regulatory mechanisms. It is more valuable to identify how a comprehensive HRV assessment reflects the degree of autonomic disorders and correlates with the severity of the disease.

We thank Reviewer2 for these crucial points.

We acknowledge the importance to avoid cyclic errors and to be more targeted about regulatory mechanisms in FM. We would like to clarify that HRV grade was a summary index developed only to globally detect the presence of ANS abnormalities so that it could be easily adopted in clinical settings to streamline decision-making (i.e., stratifying patients based on their degree of autonomic balance) and assist also healthcare providers who could be not very familiar with the interpretation of individual HRV indices (e.g., SDNN and RMSSD and total power) that are more specific. Our intention was not to consider HRV grade as a “stand-alone” index or as a substitute for standardized HRV parameters, neither to focus on the relationship between a global HRV grade and the 3 individual parameters from which it is derived – as circularity could be expected in this case. Both HRV grade and D2 were considered as complementary to linear HRV measures and not substitutes for them.

Our aim was instead to focus on the potential relationships between HRV grade (linear), D2 (nonlinear) and clinical variables, and separately, also to analyze how D2 could correlate with global and individual HRV linear and clinical variables. It is not surprising that HRV grade could show a significant correlation with single HRV linear indices. Conversely, what may add value to our work is whether HRV grade can reflect less adaptability of the autonomic system (D2) and be associated with disease impairment (e.g., FIQ), also considering that it is not always easy to summarily quantify an ongoing ANS dysfunction in the presence of multiple indices, especially when they are borderline to normative values. In our work, both HRV grade and D2 resulted moderately correlated with FIQ (r=.51 and r=-46, respectively), revealing an association between overall dysautonomia and ASN rigidity with a disease-specific index of functional impairment designed for FM patients.

Regarding analysis, we made some additions and edits in the discussion (including limitation) and conclusions (lines 261-278; 292-295; 313-316; 325-343) so that the respective role of D2 and HRV grade could be better elucidated in relation to our aims and the purpose of our analyses. We acknowledge our model requires cautious interpretations as its main focus are comprehensive indices that are system-wide and not branch-specific. Further studies could assess if using different models and measurements (e.g., specific vs. global) will be needed to determine whether such techniques could also predict the risk for severe dysregulations.

R2: The authors used the nonlinear index D2 of the HRV, which differed in patients with fibromyalgia and healthy individuals. At the same time, the studied parameter does not show the structure of autonomic disorders in patients with fibromyalgia. The authors should make a more in-depth analysis showing the structure of autonomic disorders in fibromyalgia.

Conclusion: the article needs revision, a more in-depth analysis of the HRV data for accurate clinical interpretation.

We understand Reviewer2’s concern that D2 alone cannot grasp the manifold clinical presentations and the structure of autonomic disorders in FM. However, as explained in our previous reply, our intention was not to use D2 in isolation, but rather to examine how D2 interacts with standard linear parameters (the clinical significance of which has already been identified, e.g., RMSSD as an indicator of parasympathetic tone) and clinical variables. We acknowledge that nonlinear parameters lack clinical standardization, however, their association with functional impairment (as found in our correlation with FIQ) may provide a starting point to understand the relationships between ANS balance and the specific clinical features in FM. Lower D2 values suggest a loss of dynamic complexity, consistent with a less flexible and more stereotyped autonomic response. This reduced complexity has been previously linked to maladaptive physiological states and has been observed in chronic disorders such as fibromyalgia, chronic pain, and depression. In this context, D2 may serve as a global index of autonomic system integrity, capturing aspects of nonlinear dynamics that are not evident from standard linear HRV measures. While D2 does not map directly onto a specific branch of the ANS, it reflects a holistic measure of system-level functioning, with potential implications for dysautonomia severity and chronic stress adaptation. As the clinical measures used for FM are disease-specific (e.g., FIQ), we addressed in the two different paragraphs (discussion section, lines 261-278; 311-316 these aspects, as follows:

“From correlation analysis, we also found a moderate association between D2 and an index of FM-specific impairment (FIQ). Specifically, D2 represents a nonlinear measure of system-wide complexity rather than a marker of sympathetic overactivity, while the FIQ is a self-report measure that globally assesses FM impact on a series of tasks and social activities are usually performed on weekly basis. The negative correlation between these two variables may be interpreted in view of patients’ chronic difficulties in modulating physiological responses to pain and coping with environmental challenges [39]. Maladaptive homeostatic equilibrium in chronic pain was discussed in a recent study where an asso-ciation between disease-related stress and long-term rearrangements in cardiovascular (re)activity emerged in a sample of FM patients [39]. In addition, we found significant associations between D2 and HRV linear indices (i.e., RR-Tri index, NN50, pNN50, SD2, SDNN, LF, HF, total power, VLF, and RMSSD). In particular, D2 seems to be associated with parasympathetic-dominant parameters that may result as altered in chronic pain. However, among all HRV indices considered individually, D2 was the most closely linked with FM-specific functional impairment, and this finding may suggest the utility of complementing FM standard assessment using HRV system-level measures that could reflect patients’ impairment in daily life and foster intervention targeting autonomic plasticity and resilience in FM.”

“Despite these promising results, the available data have significant limitations, including the relatively small sample size and methodological variability in the measurement of the various HRV parameters, especially when non-standardized ones are involved. In this regard, system-wise measures such as D2 and global measures such as HRV grade could complement, and not replace, standardized linear indices that are informative of branch-specific ANS dysfunctions in FM.”

Reviewer 3 Report

Comments and Suggestions for Authors

Title: Combined Proxies for Heart Rate Variability as a New Tool to Assess and Monitor Autonomic Dysregulation in Fibromyalgia and Disease-Related Impairments

The topic of this paper is undoubtedly interesting and relevant, especially given the increasing importance of heart rate variability (HRV) for the diagnosis of various diseases in recent years. The article makes significant findings, with the authors proving that the parameters used in HRV analysis, applied to the study of ECG signals in fibromyalgia (FM), have diagnostic value.

Comments:

The introduction is well structured and provides a solid background on the topic. However, in the phrase “... new composite HRV parameter...”, the definition of “new” is somewhat exaggerated. SDNN, RMSSD and total power are parameters that are part of the group of linear methods for which the European and North American Cardiology Societies gave recommendations for clinical application as early as 1996. Therefore, these three parameters, although united in a group, cannot be considered new to date, but it would nevertheless be useful to clarify what is new in the presented approach.

Materials and Methods

In the section 2.1.  Subjects, the specified age range 26-68 years is quite wide and this can lead to a problem in HRV analysis, because the age of the subjects significantly affects HRV.

The link mentioned: https://neuroclinic.thcs.it/login/login.html), where clinical data is recorded is unavailable.

In section 2.2. Neurophysiological Assessment, the formulas for the HRV parameters are known and their writing is rather redundant unless modified versions are presented.

There is no description of the figure in this section.

In section 2.3. Clinical Assessment, there is also no access to the database: https://neuroclinic.thcs.it/home.php, which calls into question the accessibility of the data. It is necessary to ensure correct access or provide an alternative.

Results

The two matrices shown in Figure 2 include parameters that are not analysed in the article. The description of the figure refers to Table 3, which is attached as supplementary material, but such a table is missing.

Discussion

The following expression: „The main reason why D2 and other non-linear parameters are rarely used in clinical practice may be due difficulty of translating mathematical measures of HRV complexity into a more intuitive neurophysiological and clinical language.“ is not entirely correct, because the main reason why nonlinear methods are rarely used in clinical practice is not related to the difficulty in interpreting mathematical measurements, but rather to the lack of standardization and the fact that these methods are still under active scientific research.

References

The Reference is not written according to the journal's requirements.

The article addresses an important topic with potential for clinical application, but requires refinement in methodology and a clearer distinction of the new approach from already existing parameters of the HRV.

Author Response

Reviewer3

R:

R3: The topic of this paper is undoubtedly interesting and relevant, especially given the increasing importance of heart rate variability (HRV) for the diagnosis of various diseases in recent years. The article makes significant findings, with the authors proving that the parameters used in HRV analysis, applied to the study of ECG signals in fibromyalgia (FM), have diagnostic value.

Comments: The introduction is well structured and provides a solid background on the topic. However, in the phrase “... new composite HRV parameter...”, the definition of “new” is somewhat exaggerated. SDNN, RMSSD and total power are parameters that are part of the group of linear methods for which the European and North American Cardiology Societies gave recommendations for clinical application as early as 1996. Therefore, these three parameters, although united in a group, cannot be considered new to date, but it would nevertheless be useful to clarify what is new in the presented approach.

We thank Reviewer3 for these comments and in particular for this key point, with which we agree. Accordingly, we changed the terminology in the title and report “new” as “global” in the body of the text. Our initial choice (“new”) only reflected the view that this global index has not yet been adopted and validated (as mentioned in the methodological section of the paper), but having derived this summary measure from well-established HRV measures, we agree that a clearer terminology is advisable. We hope the term “global” could result as more cautious and accurate, and thus prevent misunderstandings on the HRV measures collected and analyzed.

These changes may also lend support to our findings as HRV grade could contribute to enhancing screening practices for ongoing autonomic dysfunctions in patients. Indeed, this index could be used as an additional, comprehensive tool to stratify patients for screening purposes, without representing an independent measure (being already derived from standardized, specific parameters that can be used to quantify ASN abnormalities). Considering that HRV grade was associated with FM impairment, it can be useful to the clinician to get a preliminary, summary overview of patients’ clinical condition.

R3: Materials and Methods

In the section 2.1.  Subjects, the specified age range 26-68 years is quite wide and this can lead to a problem in HRV analysis, because the age of the subjects significantly affects HRV.

We agree that the specified age range is wide and could potentially impact HRV. Hence, we used age as a covariate in both the non-parametric test used for D2 comparisons, and the multivariate model (correlations). In our analyses (Paragraph 2.4), we already mentioned in the text that “Age was included as a covariate as it may influence cardiac variability” and performed the analyses accordingly.

R3: The link mentioned: https://neuroclinic.thcs.it/login/login.html ), where clinical data is recorded is unavailable.

The login link for Neuroclinic seems typed correctly and results still accessible for healthcare providers among us, however, it could possibly result inaccessible to some domains as it consists a private eHealth platform used to securely store data from neurological Italian centers. In any case, data access is possible only for authorized staff under credentials.

R3: In section 2.2. Neurophysiological Assessment, the formulas for the HRV parameters are known and their writing is rather redundant unless modified versions are presented. There is no description of the figure in this section.

We thank Reviewer3 for these comments. In the previous version of the manuscript, we decided to incorporate the original formulas of the computed HRV measurements for clarity and transparency purposes. However, as they are well-known and could burden the article readability, we have  now provided in their place Kubios link where these formulas are extensively reported and explained. The revised version (lines 153-157) states: “In particular, the values of SDNN, RMSSD, and D2 were calculated using appropriate formulas to form vectors from the time series RR, compute the number of vectors, and then determine correlation dimension (https://www.kubios.com/blog/hrv-analysis-methods/). In the software, an embedding default value of m=10 was selected.”

R3: In section 2.3. Clinical Assessment, there nclusi no access to the database: https://neuroclinic.thcs.it/home.php, which calls into question the accessibility of the data. It is necessary to ensure correct access or provide an alternative.

As per our previous comment, we specify that Neuroclinic is an active clinical system used to collect and manage data consistent with current regulations and privacy norms. As such, direct access cannot be made public as it would require specific clinical credentials to ensure patients confidentiality, also in light of the amount of different kind of clinical data stored. However, public access to the platform page could be accessible. We understand the importance of transparency and data accessibility, so we are willing to provide upon request an anonymized aggregated dataset to allow for possible verification or methodological replication.

R3: Results

The two matrices shown in Figure 2 include parameters that are not analysed in the article. The description of the figure refers to, which is attached as supplementary material, but such a table is missing.

The full list of available variables is listed in table 2 “Indices of heart rate variability and clinical variables collected”, where acronyms are also explained. For correlation analysis we considered all these variables collected from the HRV software and Neuroclinic, such as the frequency-domain HRV metrics LF, HF, and two clinical generic scales for pain that are just a complement of those used specifically for pain in FM as per Methods. The reason for inclusion is that we wanted to provide a transparent and comprehensive overview of the broader dataset structure and its internal relationships. Moreover, we included Table 3 (which is consistent with Figure 2) and its caption in the body of text, following the references, as we noticed it is missing in the current version of the manuscript under revision. An additional link to supplementary materials is included in the appropriate section of the manuscript.

R3: Discussion

The following expression: „The main reason why D2 and other non-linear parameters are rarely used in clinical practice may be due difficulty of translating mathematical measures of HRV complexity into a more intuitive neurophysiological and clinical language.“ is not entirely correct, because the main reason why nonlinear methods are rarely used in clinical practice is not related to the difficulty in interpreting mathematical measurements, but rather to the lack of standardization and the fact that these methods are still under active scientific research.

We appreciate this clarification and fully acknowledge that a primary limitation to their broader use lies in the current lack of standardization which makes it difficult to interpret whether a given value reflects pathological autonomic function in individual patients. Without well-established population benchmarks, their integration into clinical practice remains challenging. This point not only does not conflict with our interpretation, but enriches it because it highlights the need for standardization of these techniques and their consequent exstablishment in the clinical workflows. In addition to this key point, in clinical settings where research on HRV is ongoing there is still a tendency to favor linear parameters because they are deemed more intuitively translatable in terms of clinical implications. Accordingly, we edited the Discussion (lines 243-247) as follows: “The limited use of D2 and other nonlinear HRV parameters in clinical settings may primarily depend on the lack of their standardization. In addition, the difficulty of translating mathematical measures of HRV complexity into a more intuitive neurophysiological and clinical language may represent a further challenge to D2 adoption in clinical settings.”

Overall, we believe nonlinear parameters like D2 may offer valuable physiological insights, especially in the context of conditions such as chronic pain and FM, where autonomic dysregulation is a known contributor. A reduced system complexity, as indicated by lower D2 values, may reflect a state of impaired autonomic adaptability and potentially indicate more pronounced dysautonomia. Along with HRV grade, this might represent a starting point for stratifying disease severity and/or treatment responsiveness. We better addressed this point in the Conclusions, which were rearranged and expanded, lines 324-344 . We hope this perspective contributes to a broader discussion on the clinical utility of HRV complexity measures, and we believe future research aimed at standardizing and validating these metrics will help unlock their full potential in translational medicine.

R3: References

The Reference is not written according to the journal’s requirements.

We thank Reviewer 3 for this suggestion. Double numbering error in the endnote bibliography, which has been removed.

The article addresses an important topic with potential for clinical application, but requires refinement in methodology and a clearer distinction of the new approach from already existing parameters of the HRV.

We thank the Reviewers for their work that gave us the opportunity to make changes and amendments to improve the overall quality of our manuscript.

Round 2

Reviewer 2 Report

Comments and Suggestions for Authors

the manuscript has been sufficiently improved to warrant publication in Sensors

Reviewer 3 Report

Comments and Suggestions for Authors

The article looks better after incorporating the requested corrections.